# Ergonomics and Occupational Health: Knowledge, Attitudes and Practices of Nurses in a Tertiary Hospital in Botswana

**DOI:** 10.3390/healthcare13010083

**Published:** 2025-01-06

**Authors:** Kagiso Kgakge, Paul Kiprono Chelule, Themba Geoffrey Ginindza

**Affiliations:** 1Discipline of Public Health Medicine, School of Nursing & Public Health, University of KwaZulu-Natal, Durban 4041, South Africa; ginindza@ukzn.ac.za; 2Boitekanelo College, Tlokweng P.O. Box 203156, Botswana; 3Department of Public Health, School of Healthcare Sciences, Sefako Makgatho Health Sciences University, Pretoria 0208, South Africa; paul.chelule@smu.ac.za; 4Cancer & Infectious Diseases Epidemiology Research Unit (CIDERU), College of Health Sciences, University of KwaZulu-Natal, Durban 4041, South Africa

**Keywords:** ergonomics, musculoskeletal disorders, lower back pain, occupational health nursing, Botswana

## Abstract

**Background**: Musculoskeletal disorders (MSD) are, to this day, considered one of the major occupational health risks, especially among healthcare workers. Poor working conditions, such as awkward postures, are associated with the development of MSD. This study aimed to evaluate the knowledge, attitudes, and practices (KAP) of nurses at a public tertiary hospital in Botswana relating to ergonomic principles. **Methods:** The researchers conducted a cross-sectional survey, using a self-administered questionnaire to collect the data. The researchers employed Stata v18 (StataCorp, USA) to perform descriptive and inferential statistics. The chi-square test was used to determine the association between knowledge levels and sociodemographic variables. *p*-values ≤ 0.05 were deemed statistically significant. **Results**: In total, 306 nurses participated in the study, and a response rate of 88.4% was achieved. The mean age was 35.5 (SD± 8.79) years. Most (69%) participants were female nurses. About 99.3% (95%CI: 97.7–99.9) of the participants were familiar with the concept of ergonomics. Only a small proportion of participants (26%) possessed high levels of knowledge, exhibited positive attitudes, and demonstrated good practices in relation to ergonomic principles, following the composite analysis. A statistically significant relationship was found to exist between sex and practice (*p* = 0.030) and between length of work and practice (*p* = 0.013). **Conclusions**: The KAP analysis indicated that most nurses had poor practices regarding essential ergonomic principles. These findings could inform policy development and enable employers to design prevention strategies, especially those aimed at preventing lower back pain (LBP).

## 1. Introduction

Numerous work-related factors pose a threat to employees’ health and safety in almost every workplace [1,2] This is especially true for medical personnel in clinical and hospital settings, where prolonged exposure to physical demands is high. The International Labour Organisation (ILO) estimates that 2.3 million workers die following occupational accidents and work-related diseases [1]. MSD is defined as injuries and disorders that happen in relation to work, affecting muscles, tendons, ligaments, joints, peripheral nerves, and supporting blood vessels with consequent pain and discomfort. Injuries associated with MSD include disc prolapse, spinal disc degeneration, muscle tears, and spinal fractures [3,4].

MSD is reported to have the highest prevalence of occupational health diseases, affecting 1.71 billion people. Lower back pain (LBP) refers to the pain in the lumbosacral area of the spine encompassing the distance from the 1st lumbar vertebra to the 1st sacral vertebra. The most frequent site of LBP is in the 4th and 5th lumbar segment. It can be either acute, sub-acute, or chronic and affects 568 million individuals [5]. Healthcare professionals, including nurses, are continuously exposed to the biological, chemical, physical, social, or mechanical workplace hazards. Long-term exposure to these hazards has been shown to have detrimental effects on the health of those exposed [6,7]. It is, therefore, crucial to have greater insights into ergonomic principles in the workplace to sustain an acceptable work performance level.

Ergonomics is the design of the workplace, machines, equipment, tools, environment, and how work activities are executed, considering humans’ physical, psychological, physiological, and biomechanical capabilities. It optimises the productivity of work systems while ensuring employee safety and wellness [8]. Nurses often lack the knowledge of ergonomic principles; this leads to health risks [9] such as MSD and LBP [10,11], due to the manual handling of obese patients, awkward postures, long hours of work, and repetitive tasks with the consequent strenuous effect on the spine [12,13]. Despite the effectiveness of ergonomic principles in preventing injuries [14], awareness and training among nurses is insufficient [15,16], leading to a high prevalence of MSD, ranging from 33% to 90.1% [3,17,18,19,20,21], and LBP [22,23,24,25].

A study conducted by Saremi et al. (2013) [10] and Juibari et al. (2010) [7] in Iran documented a sound level of knowledge among nurses. The healthcare workers in that context were forced by their poor work environments to develop an awareness of ergonomic issues [7], thusergonomics can serve as a predictor of MSD risk [26]. The findings of studies conducted elsewhere revealed that there is a strong association between knowledge of ergonomic principles and the reduction of MSD in terms of being conversant in the techniques for transporting patients [16,27].

Unlike professionals in other fields, nurses are exposed to more occupational hazards. Negligence in this regard could result in a decline in the standards of quality nursing care [7]. Several studies have documented the consequences of LBP among nurses; for example, [28] reported that LBP accounts for 750,000 lost days per year. On the other hand, employers in the United Kingdom lost an estimated 6.9 million workdays due to employees taking sick leaves and being absent from work [29]. Also, activity limitation and economic burdens among individuals, families, and communities have been reported, as well as workplace injury compensation and treatment costs [30]. These factors have an impact on the quality of life for nurses and the quality of nursing care that they provide [7].

Various studies globally, including those in Botswana, have largely investigated the prevalence and associated risk factors of MSD among nurses, and one study conducted by Kgakge et al. [31], revealed a high prevalence of LBP (78%) among nurses. However, data on the knowledge, attitudes, and practices of nurses regarding ergonomic principles remain scarce [4,23,31,32,33,34,35,36]. To the best of the researchers’ knowledge, KAP studies in Africa have provided limited data on ergonomic principles. For instance, Aluko et al. (2016) assessed the workplace hazards and safety practices among selected health care workers (HCWs) in Nigeria. The findings revealed that most (89%) respondents were knowledgeable about workplace hazards [37]. Epidemiological data on such studies are crucial to provide insights into ergonomic principles and the associated risk factors to guide the development of successful prevention strategies in various countries. In response to the Global Strategy on Occupational Safety, which is supported by the Promotional Framework for Occupational Safety and Health Convention (No. 187), workplaces must promote healthy working environments [1]. Therefore, this study aimed to evaluate the knowledge, attitudes, and practices of nurses towards ergonomic principles in public tertiary hospitals in Botswana.

## 2. Materials and Methods

### 2.1. Study Design

The researchers conducted a hospital-based analytic cross-sectional study among nurses at Princess Marina Hospital (PMH), a tertiary public hospital in Botswana. This study was conducted in compliance with the Declaration of Helsinki [38]. The report was prepared according to the guidelines of Strengthening the Reporting of Observational Studies in Epidemiology (STROBE) [39].

### 2.2. Study Setting and Population

This single-centre study was conducted between March and April 2023 at Princess Marina Hospital (PMH), the largest tertiary and referral hospital, with a capacity of 500 beds. The hospital offers a wide range of inpatient and outpatient services. Botswana’s public healthcare system is organised into the referral health system, district hospitals, and primary hospitals and offers universal healthcare to all its citizens.

### 2.3. Participants and Eligibility Criteria

The study sample consisted of nurses who were willing to participate and had signed the informed consent form. Nurses working in different wards of PMH and frequently involved in the manual handling of patients were also included in the study.

### 2.4. Sampling, Sample Size, and Recruitment Strategy

The researchers calculated the sample size, using the Yamane formula (1967), which provides a simplified formula to calculate sample sizes. A 95% confidence level and a *p*-value of 0.05 were assumed for the calculation. In this formula, *n* is the sample size, *N* is the population size of nurses, which is 549, and *e* is the level of precision. The applied formula yielded
$$n=\frac{N}{1+N{\left(e\right)}^{2}}\;=549/(1+549\times (0.05\times 0.05)=231.$$

Therefore, the sample size calculated for this study was 231, plus an additional 35 (15%) to account for a non-response rate and a design effect of 1.3 (80), resulting in a final sample size of 346.

After obtaining a duty roster for nurses working in various wards in PMH, systematic random sampling was used to select nurses from each ward to achieve the desired sample of 346. Nurses were approached individually during off-peak hours to avoid interruption of patient care. Details of the study were then further explained to the participants.

### 2.5. Study Instruments

A self-administered, standardised structured questionnaire was designed using an informational booklet from the Occupational Safety and Health Training guide (2018), and the related literature from similar studies [15,16,37,40]. Thereafter, it was distributed among participants to collect the socio-demographic data, knowledge level, attitudes, and practices of nurses relating to ergonomic principles of musculoskeletal conditions. The researchers took a day to pilot the questionnaires among participants at a separate site, the Scottish Livingstone Hospital (SLH) in Botswana, which did not form part of the study sites. A total of 35 questionnaires were piloted. The purpose of the pilot study was to test the questionnaire and the proposed recruitment methods. The pilot study was also conducted to test the readability of the study consent form and to see if nurses perceived the study as a threat. The researcher ensured the validity of the questionnaire by consulting supervisors, who are expert leads in the field of occupational health.

### 2.6. Data Collection and Data Management

All participants willing to participate in the study signed the informed consent forms after detailed information on the study was shared with them. A structured, standardised questionnaire was administered by trained research assistants to collect data on the nurses’ demographic variables, knowledge levels, attitudes, and practices regarding ergonomic principles. The nurses who were willing to participate in the study took the questionnaires to fill in during their free time. The research assistants later collected the complete questionnaires. There were no repeated measures or follow-ups as the data were collected in a single instance. The researchers entered and stored the data in Stata v18. Each participant was assigned a unique study identity number that would link the questionnaire with the electronic database.

### 2.7. Data Analysis

The researchers used Stata 13.0SE (Stata Corp., College Station, TX, USA) to process and analyse the data. Before analysis, the data were checked for errors and missing values. Participants’ demographic characteristics were summarised using descriptive statistics, with categorical variables presented as frequencies and percentages, and continuous variables reported as means and standard deviations (SD). Data visualisation in relation to nurses’ composite knowledge, attitudes, and practices of ergonomic principles was performed using a pie chart. Tests for association between knowledge, attitudes and perceptions, and respondent demographic characteristics were conducted using chi-square tests. Statistical significance was set at *p* < 0.05

### 2.8. Categorising the Total KAP Scores

Based on the nurses’ knowledge, attitudes, and perceptions of occupational hazards and safety practices in a related study on Nigerian healthcare workers, data for our study on knowledge questions on ergonomics were categorised into “knowledgeable” and “not knowledgeable”, based on the answers provided to each question. Practice and attitude questions were categorised into “agree”, “neutral”, and “disagree”. Composite knowledge, attitude, and practice scores were derived by summing the scores for each attribute. For the knowledge composite score, a score below the median was classified as “low”, and a score equal to the median or greater was classified as “high”. Similarly, for attitude, a score below the median was classified as “negative”, and greater than or equal to the median was “positive”. For practice, a score below the median was classified “poor”, while a score greater than or equal to the median was classified as “good”. A KAP composite score for knowledge, attitude, and practice with eight categories was created, based on the categories of knowledge (1 = high, 2 = low), attitude (1 = good, 2 = poor), and practice (1 = positive, 2 = negative). The KAP composite categories were as follows: 1 = high, positive, good (1, 1, 1), 2 = low, negative, poor (2, 2, 2), 3 = high, positive, poor (1, 1, 2), 4 = high, negative, poor (1, 2, 2), 5 = high, negative, good (1, 2, 1), 6 = low, positive, good (2, 1, 1), 7 = low, negative, good (2, 2, 1), and 8 = low, positive, poor (2, 1, 2).

### 2.9. Ethical Consideration

The researchers obtained approval to conduct the study from the Botswana Ministry of Health, under the Health Research and Development Division (HPRD: 6/14/1), and the University of KwaZulu-Natal Biomedical Research Ethics Committee (BREC) (BREC/00004365/2022). Permission to conduct the study was obtained from the Princess Marina Hospital Ethics Committee and its management. Participation in the study was voluntary, and all participants gave their informed consent before the data collection began.

## 3. Results

### 3.1. Characteristics of the Study Population

Table 1 shows the socio-demographic characteristics of the study participants. The study was conducted between March and April 2023. Of the 346 participants that were recruited, only 306 (88.4% response rate) completed and returned the questionnaires successfully. About 69.3% (n = 212) of participants were female, with a mean (±standard deviation [SD]) age of 35.5 (±8.79) years. Most (84.3%) of the study participants had a Diploma qualification in Nursing, with less than 10 years’ work experience (50%), and were single (62.7%).

### 3.2. Knowledge, Attitudes, and Practices of Nurses Towards Ergonomic Principles

Table 2 illustrates the participants’ knowledge of ergonomic principles. Nearly all the respondents (304, 99.3%, 95%CI: 97.7–99.9) had an idea of what ergonomics is. In relation to ergonomic risk factors, most of the nurses (248, 78.8%, 95%CI: 73.7–83.2) were knowledgeable about ergonomic factors, as they indicated that all the listed factors were correct. Over 97% of the participants knew that ergonomic risk factors and injuries impact employers and employees directly and indirectly. In total, 142 (46.4%, 95%CI: 40.7–52.2) of the participants indicated that they were not aware that twisting their back while lifting objects is most likely to expose the lower back to a greater risk of injury. Regarding the symptoms of MSD (pain, numbness, or tingling sensation in the affected areas), almost all participants (303, 99%, 95%CI: 97.2–99.8) indicated that they were aware of ergonomic risk factors.

About 89.9% (95%CI: 85.9–93.0) believed that age was a contributing factor to LBP, and 90.5% (95%CI: 86.7–93.6) believed that excessive body weight can cause LBP. Most of the participants (37.3%, 95%CI: 31.8–42.9) were still undecided on whether smoking plays a role in the development of LBP. A higher proportion of the participants did not consider education and marital status to be ergonomic risk factors for LBP, with 44.4% (95%CI: 38.8–50.2) and 68.3% (95%CI: 62.3–73.4), respectively.

In total, 15 items were used to assess the nurses’ practices, based on participants’ self-reported practices during work, in relation to environmental, organisational, and psychological factors. Almost all the participants reported that they performed heavy manual lifting (of patients) from floors to beds and lifted and moved equipment in their workplace, with 99.7% (95%CI: 98.2–100.0) and 97.4% (95%CI: 94.9–98.9), respectively. About 94.1% (95%CI: 90.9–96.5) of the participants reported positioning patients in beds, while over two-thirds of the participants (97.3%, 95%CI: 94.9–98.9%) reported standing for long hours during the execution of their nursing duties. A total of 99% of the participants stated that they bend while working, and 96.1% (95%CI: 93.3–98.0) reported exerting force when working. Most participants (93.1%, 95%CI: 89.7–95.7) reported they were physically unfit, while 99.4% reported work overload. Most reported that they had experienced work pressure and working in poor environments, with 96.7% (95%CI: 94.1–98.4) and 95.4% (95%CI: 92.4–97.5), respectively. A total of 60.1% (95%CI: 54.4–65.7) of the participants reportedly received no support from managers/management. In addition, participants reported that there was work control, lack of job satisfaction, and work fatigue and that they were emotionally distressed, with 60% (95%CI: 54.4–65.7), 56.2% (95%CI: 50.4–61.8), 90.5% (95%CI: 86.7–93.6), and 80.4% (95%CI: 75.5–84.7), respectively (Table 2).

### 3.3. Composite Rating Scores for Knowledge, Attitude, and Practice

Based on the composite KAP score, about 26% (95%CI: 21.3–31.4) of participants demonstrated high knowledge levels, positive attitudes, and good practice. In total, 17% (95%CI: 13–21.7) of the participants demonstrated high knowledge levels, negative attitudes, and poor practice of ergonomic principles, while 15% (95%CI: 11.2–19.5) displayed high knowledge levels, negative attitudes, and poor practice. About 10% (95%CI: 7.0–14.1) of the participants displayed low knowledge levels, negative attitudes, and poor practice, while 10% (95%CI: 7.0–14.1) of the participants demonstrated high knowledge levels, positive attitudes, and poor practice (Figure 1).

### 3.4. Risk Factors Associated with Knowledge, Attitudes, and Practices

A statistically significant relationship was found to exist between gender and practice (χ^2^(df = 2) = 4.73, *p* = 0.030), while there was no association between gender and knowledge (χ^2^(df = 2) = 1.19, *p* = 0.276) and no association with attitude (χ^2^(df = 2) = 0.350, *p* = 0.554). The results further revealed an association between the length of work and practice (χ^2^(df = 2) = 8.70, *p* = 0.013). A significant difference was not found between length of work and knowledge (χ^2^(df = 2) = 2.42, *p* = 0.298), as well as attitude (χ^2^(df = 2) = 4.08, *p* = 0.130). In addition, age was not significantly associated with knowledge (χ^2^(df = 2) = 2.00, *p* = 0.368), attitude (χ^2^(df = 2) = 4.92, *p* = 0.085), and practice (χ^2^(df = 2) = 3.65, *p* = 0.161). The results further show no relationship between marital status and knowledge (*p* = 0.275), attitude (*p* = 0.195), and practice (*p* = 0.634). Lastly, education was not significantly associated with knowledge (*p* = 0.824), attitude (*p* = 0.062), and practice (*p* = 0.322). The details of the findings are depicted in Table 3.

## 4. Discussion

This study is, to the researchers’ knowledge, the first study conducted to assess the knowledge, attitudes, and practices of ergonomics principles among nurses working in a public hospital in Botswana. The study focused on demographic variables, the nurses’ knowledge of ergonomics, and the assessment of attitudes and practices of ergonomics in the workplace.

### 4.1. Demographic Characteristics of Participants

Most of the study participants were predominantly female. Nursing is by and large a female profession, a finding consistent with most other studies [7,15,37], but an exception is noted in one study conducted in Saudi Arabia [41]. The inconsistent finding could be linked to the Saudisation programs’ non-discriminatory approach to employment. Another reason is that the nursing profession in Saudi Arabia has historically been dominated by males rather than females. This is due to a combination of cultural, social, and economic factors. Most of the participants in this study had less than 10 years’ work experience; this is consistent with the findings of studies by Saad and Ebraheim, as well as Ali et al. [6,40]. The researchers had anticipated this observation because the mean age of respondents was 35 years, like the mean age of participants in studies conducted elsewhere [10,15]. Most of the participants in this study had a Diploma qualification, unlike participants in the study conducted by Aluko et al. (2016) [37] who were mostly in possession of a bachelor’s degree [37]. This is probably because most nursing training institutions in Botswana offer Diplomas in Nursing. This study reports that most of the participants were single, as observed in a similar study conducted by Saremi et al. [10], but were less in other studies [6,15,37].

### 4.2. Nurses’ Knowledge of Ergonomics Principles

The researchers used 15 items to assess the nurses’ knowledge levels of ergonomic principles. In general, results revealed that nurses were more knowledgeable about ergonomics principles in the workplace. These findings are consistent with the findings of a similar cross-sectional study by Juibari et al. [7], but inconsistent with the findings of studies by [16,42], conducted elsewhere, where nurses did not have an awareness and did not receive training on ergonomic principles. A high proportion of nurses were familiar with the concept of ergonomics: only a handful were not familiar. Also, 99% of the participants stated that symptoms of MSD include pain, numbness, or a tingling sensation in the affected area. These findings are not in line with the findings of other studies that revealed low levels of ergonomic knowledge among nurses [16,42]. Although nurses in this study demonstrated a high level of knowledge, a previous study on these participants revealed that LBP was mostly prevalent among 78% of the nurses, which could probably mean poor practice of ergonomic principles among nurses.

This study assessed the nurses’ knowledge of different ergonomic risk factors.

All the options given were correct. However, some participants did not know that all the given options were ergonomic risk factors. Evidence from other published studies suggests that some risk factors, like lifting and transferring heavy patients, an activity that the nurses engage in repetitively, require force while maintaining uncomfortable postures, exposing them to MSD such as LBP [4,20,22,31]. Furthermore, poor working environments such as lack of access to lifting devices expose nurses to a high risk of MSD [19,21,22,43,44].

A large proportion of the nurses further indicated that bending at the lower back while working increased the likelihood of LBP occurrence. Nurses assumed uncomfortable positions in their day-to-day nursing activities as they executed their duties such as the manual lifting of patients, bed baths, bed making, wound dressing, and adjusting heights of beds, to mention a few, and these activities forced them to assume awkward postures [4,20,44,45]. Furthermore, most of the nurses indicated that standing, sitting, or otherwise remaining in one posture for a long duration while performing a task could increase the likelihood of MSD injury.

Participants were also assessed on different preventive strategies used to prevent ergonomic injuries. The findings of this study suggest that the nurses who participated in this study are more knowledgeable about these strategies (97.7%). This is indicated by the fact that the nurses use devices like hoists to lift patients, which is deemed the most effective strategy. The findings of this study are inconsistent with the findings of a study by Alhazim et al. (2022) [41] who reported that the participants had a fair knowledge of strategies to prevent ergonomic injuries [41]. Different studies suggest different methods and techniques that nurses can use to control MSD in the workplace, such as using assistive devices to lift heavy patients to reduce the prevalence of MSD [22,46,47]. Although the nurses are aware of the need to use these devices, they are limited by their work environments, as such devices are not available. Hence, they are consistently exposed to MSD injuries.

Even though most of the participants chose the answer option, “use of devices is a preventive strategy”, all the options were correct. However, only 87.6% were aware that all the given answers were correct. Other options included the rotation of workers in respect of different tasks and the use of personal protective equipment (PPE). It is vital that nurses are rotated in the workplace because the workload is not the same for different wards; hence, this can help minimise exposure of individual nurses to the same tasks that can expose them to MSD injuries. Moreover, PPE is believed to provide some barrier protection between the worker and the hazards, although evidence of its effectiveness remains inconclusive (The National Institute for Occupational Safety and Health (NIOSH)).

Almost all participants perceived effective lifting techniques as one strategy that can be used to reduce the prevalence of MSD. Poor lifting techniques of heavy patients are greatly implicated in the development of MSD among nurses as it normally results in spinal overload. Systematic reviews and meta-analyses have documented evidence on determinants of LBP risk factors such as heavy workloads that include patient manual handling [48,49]. A higher proportion of nurses believed that education and awareness of ergonomics can reduce the prevalence of MSD. Educational programmes on ergonomics play a crucial role in reducing the prevalence of MSD [50] among nurses. However, this study does not provide evidence of the implementation of such programmes in participants’ work settings. In this study, 96.4% of the nurses believed that stretching exercises have a positive effect on the reduction of MSD. Physical therapy has been widely suggested in several studies as one preventive strategy for MSD injuries [51,52].

The effects of MSD, due to factors such as poor work ergonomics among nurses, have been reported in studies conducted elsewhere. The findings of this study show that most nurses (97.7%) agree that ergonomic risk factors and injuries attract direct and indirect costs. Likewise, the findings of this study are in line with other published studies on the consequences of MSD such as increased costs of seeking treatments and the consequent compensations associated with injuries related to MSD [21,25,53]. Similarly, 95.4% of the nurses believed that reduced work productivity, lost workdays, and temporary or permanent disability were linked to MSD injuries. This finding is corroborated by the findings of several studies [35,52,54,55]. Lastly, 92.8% of the participants agree that to create a healthy environment that reduces the prevalence of ergonomic injuries, management should let end users participate in the selection of equipment in the workplace.

### 4.3. Assessment of Attitudes on Demographic Characteristics in Relation to Ergonomic Principles

Some demographic variables have been strongly associated with ergonomics. This study assessed the attitudes of nurses on personal factors in relation to ergonomics that can contribute to LBP. Only five items were used. Most of the participants perceived age as a predictor of LBP. Age plays a role in ergonomic risk and work performance among nurses. The link between age and LBP has been shown in previous studies [18,56]. Likewise, a higher proportion of the participants in this study concurred with the findings of previous studies. Some studies have reported a higher prevalence of LBP among the younger nurses, which has been linked to a lack or limited knowledge on ergonomic principles, such as the safe manual handling procedures of patients. LBP prevalence among the younger nurses is also due to the fact that they tend to be allocated more duties in their wards compared to the elderly nurses. On the contrary, some studies found that older nurses suffer more from LBP, due to the long duration of exposure to risk factors in their workplace [21,57,58].

More than three-quarters of participants in this study perceived body weight as a risk factor for LBP. This finding is in line with the study by Cilliers and Maart [59]. While some studies found a link between the body weight of nurses and LBP [60], other studies found no link between the two variables [61]. The World Health Organisation (WHO) defines “overweight” and “obesity” as “abnormal or excessive fat accumulation that may impair health” [62]. These conditions are commonly associated with back pain. It is crucial for nurses to be aware of the dangers of being overweight and obese. There is evidence that being overweight and obese have become more common among nurses, with consequent negative health effects, missed work, poor productivity, and cost implications for treatment [63].

Most of the participants in this study did not perceive smoking as a personal risk factor that can contribute to the development of LBP. The findings are consistent with the findings of a similar study conducted in South Africa [59]. However, various studies [64] from elsewhere have documented the relationship between smoking and the development of LBP. This association can be explained by decreased blood flow to spinal structures, changes in the nutrition that the intervertebral discs receive, and the neuroendocrine effects of nicotine [65]. It is worrisome that most of the participants were not aware of this association, as nurses might continue to be exposed to the risk of smoking due to ignorance in that regard.

Almost three-quarters of the participants did not believe marital status to be a risk factor for LBP. Some studies have found a significant association between marital status and LBP, especially among female nurses. This is because married women carry a large proportion of household chores, due to their anatomical and physiological structures. This implies that socio-demographic factors may or may not predict the occurrence of LBP among nurses.

### 4.4. Practices Among Nurses in Relation to Ergonomic Principles Contributing to Low Back Pain

Physical ergonomics is perceived as a contributing factor to the development of MSD such as LBP [1,8,25] among nurses. Fifteen items were used to assess the practices of nurses at the Princess Marina Hospital. Almost all participants in this study were involved in the heavy manual lifting of patients on their day-to-day nursing duties. This, in the long run, strains their back and causes injury. There is evidence of a link between the manual handling of patients and the development of LBP, which is further linked to a high prevalence of MSD among nurses [10,66]. Almost all the nurses who participated in this study indicated that they adopt awkward postures when lifting patients and exert force when executing their nursing duties. This might be due to a lack of proper lifting devices and techniques. These activities lead to increased stress on the lumbar spine and strain the back muscles and spinal structures.

Poor environments include issues such as lack of access to devices that can be used to assist the nurses in lifting patients, lack of motivation, support from supervisors, shortage of staff, long working hours, lack of training on ergonomics, and working in certain high care wards, like the intensive care units [19,35,43,44]. All these environmental risk factors ultimately lead to fatigue and emotional distress as most participants in this study alluded to. Stress and anxiety are the main psychosocial factors that may cause LBP. This is due to job obligations that require nurses to work night shifts, monotonous work life, job dissatisfaction, un-supportive work culture, and a lack of rest at the workplace. Consequently, all these affect the quality of patient care. However, the findings of one study conducted in South Africa contradict this finding: only 15% of the participants were of the view that psychological distress could cause LBP [59]. This apparent lack of awareness suggests poor recognition of hazards in the workplace, as well as poor practice of ergonomic principles. A study conducted in China did not find any association between LBP and psychological distress [67]. The onset of pain, due to physical injuries, can trigger a chronic dysfunction of physiological and psychological central nervous systems, resulting in chronic pain [68].

### 4.5. Comparative Analysis of Knowledge, Attitudes, and Practices of Participants

It is crucial for nurses to be knowledgeable on ergonomic principles to prevent workplace injuries. Being aware of lifting techniques, proper body mechanisms, and workstation configurations can improve the practices of these nurses. The composite analysis pertaining to this study showed that only 26% of the participants were more knowledgeable, displayed positive attitudes, and had good practices. These findings were comparatively lower than the 38% observed in a study conducted by Aluko et al. in Nigeria [37]. Surprisingly, when assessing the knowledge aspect of the nurses in this study, high scores were obtained. This shows that the nurses might be knowledgeable about these ergonomic principles but are not implementing ergonomic principles in their daily routines. The remaining composite analysis pertaining to this study showed that a call for action is required to improve the status quo. In addition, positive attitudes influence the implementation of good practices, such as adherence to ergonomic measures like safe lifting techniques, maintaining proper posture, using ergonomic equipment, and taking breaks. In this regard, the researcher is of the view that the Health Belief Model (HBM) postulates that behaviour changes when people see a health risk such as MSD and think that adopting certain behaviours like using safe lifting techniques will lessen that risk. The nurses’ knowledge of the dangers of improper ergonomics can influence their behaviour.

This study has demonstrated a relationship between gender and practice, in agreement with the findings of Juibari [7], while another similar KAP study found that gender was significantly associated with knowledge [37]. This means that the gender of participants influenced the practice of ergonomic principles. However, participants’ knowledge and attitudes were not influenced by gender. Furthermore, the length of work was shown to be associated with practice. This finding was anticipated, as it is believed that the more years of experience, the better the practice; hence, in most cases, MSD prevalence is more inclined towards younger nurses [44,69,70], who still do not have experience in issues of practice, such as patient manual handling and proper techniques.

### 4.6. Strengths and Limitations

To our knowledge, this is the first study to evaluate the KAP of ergonomic principles among nurses in Botswana, and the results of this study are useful for future research use. In addition, the strength of this study is its focused approach within a single hospital, which allowed for a detailed and in-depth exploration of the specific population in that context. The limitation of this study is that it was cross-sectional; therefore, it cannot prove causality, but rather associations. In addition, this study was conducted in one hospital; hence, the results cannot be generalised. Furthermore, self-reported data are difficult to control for recall bias. Lastly, participants might have overestimated their knowledge or positive attitudes due to social desirability bias. Thus, large-scale studies are recommended that involve many hospitals to give more accurate estimates and to generalise the findings and also future research using longitudinal studies to explore how targeted interventions impact the general KAP of nurses.

## 5. Conclusions

This study aimed to assess nurses’ (KAP) regarding ergonomic principles in the workplace. Using a composite KAP analysis, the study revealed that slightly over a quarter of the participants demonstrated higher levels of knowledge, positive attitudes, and good practices related to ergonomics. However, the remaining participants fell into categories that require targeted interventions, such as educational programs, positive reinforcement, and behaviour change model-based strategies. These findings suggest that offering structured training programs designed to improve nurses’ knowledge, skills, and behaviours can significantly contribute to creating safer and healthier work environments. The social and policy implications of this study are considerable. It underscores the importance of incorporating ergonomic training into standard nursing education, strategic partnerships between academic institutions and industries to address the gap between theory and practice, and providing ongoing professional development to enhance workplace safety and reduce injury rates. Periodic training programs on safe lifting techniques and the implementation of organisational policies can support the wellness of employees to improve job satisfaction.

## Figures and Tables

**Figure 1 healthcare-13-00083-f001:**
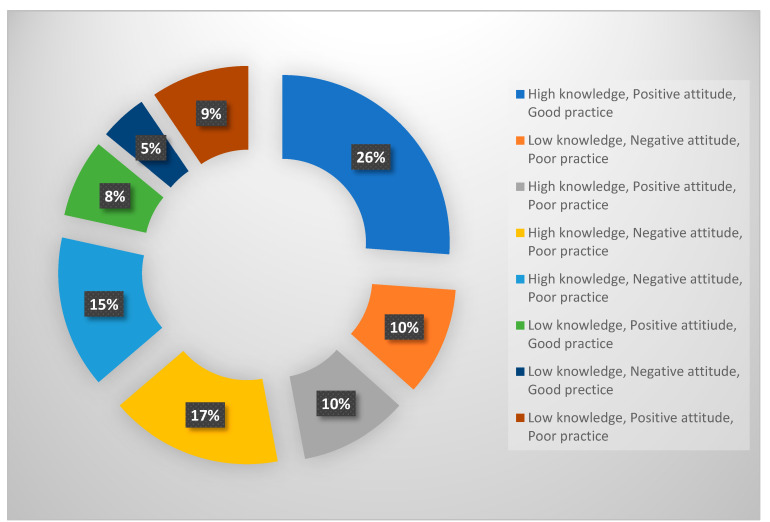
Composite knowledge, attitudes, and practices of ergonomic principles in the workplace among nurses.

**Table 1 healthcare-13-00083-t001:** Demographic variables of participants (n = 306).

	Frequency	Percentage (%)
Age		
Mean (SD) = 35.53 ± 8.79		
<30	94	30.7
30–40	146	47.7
>40	66	21.6
Gender		
Male	94	30.7
Female	212	69.3
Education level		
Diploma	258	84.3
Degree	45	14.7
Masters	3	1
Marital status		
Single	192	62.7
Married	114	37.3
Ward		
Male orthopaedic	19	6.2
Antenatal	11	3.6
Gynaecological	14	4.6
Labour	19	6.2
Oncology	14	4.6
Private	7	2.3
Male medical	18	5.9
Paediatric surgical	14	4.6
Paediatric medical	14	4.6
Spinalis	9	2.9
Female medical ward	15	4.9
A&E	10	3.3
ICU	22	7.2
Postnatal	22	7.2
Female orthopaedic	11	3.6
NNU	25	8.2
Female surgical	17	5.6
Male surgical	27	8.8
Main orthopaedic	18	5.9
How long have you been a nurse in years?		
Duration range 1–41; Mean (SD) = 11.79 ± 8.60		
1–10	153	50
11–20	106	34.6
21–30	34	11.1
31–40	12	3.9
41–50	1	0.3

**Table 2 healthcare-13-00083-t002:** KAP towards ergonomic principles among participants (n = 306).

Description of Knowledge of Ergonomics	Knowledgeable	Not Knowledgeable	
	n (%) (95%CI)	n (%)	
Ergonomics is the study of the “fit” between the physical demands of the workplace and the employees who perform the work.	304 (99.4)	2 (0.7)	
Ergonomic risk factors include vibration, repetition, environment, force, posture, work organisation, and contact stress.	241 (78.8)	65 (21.2)	
Ergonomic risk factors and injuries result in direct and indirect costs for employers and employees.	299 (97.7)	7 (2.3)	
How can employers prevent/reduce ergonomic hazards?	268 (87.6)	38 (12.4)	
Repetition is a measure of how frequently we complete the same motion or exertion during a task.	286 (93.5)	20 (6.5)	
Bending at the lower back while working increases the likelihood of lower back disorders.	297 (97.1)	9 (2.9)	
Which movement will most likely expose the lower back to a greater risk of injury? (Twisting while lifting objects)	164 (53.6)	142 (56.4)	
To create a healthy work environment that reduces the risk of ergonomic injuries, management should let end users participate in the selection of equipment in the workplace.	284 (92.8)	22 (7.2)	
Effective lifting techniques can reduce the prevalence of musculoskeletal disorders.	304 (99.4)	2 (0.7)	
Standing, sitting, or otherwise remaining in one posture for a long duration while you perform a task can increase the likelihood of injury.	294 (96.1)	12 (3.9)	
Stretching exercises do have a positive effect on the reduction of musculoskeletal disorders.	295 (96.4)	11 (3.6)	
Education and awareness of ergonomics reduce the prevalence of musculoskeletal disorders.	299 (97.7)	7 (2.3)	
Symptoms of musculoskeletal disorders include pain, numbness, or a tingling sensation in the affected areas.	303 (99.0)	3 (1.0)	
Reduced work productivity, lost workdays, temporary or permanent disability are consequent results of musculoskeletal disorders.	292 (95.4)	14 (4.6)	
Ergonomic risk factors include tools, equipment, furniture, machinery, materials, and workstation layout.	285 (93.1)	21 (6.9)	
Attitudes on personal ergonomic factors contributing to LBP	Agree	Undecided	Disagree
	N (%)	N (%)	N (%)
Age has an influence on the development of low back pain.	275 (89.9)	13 (4.3)	18 (5.9)
Body weight can lead to low back pain.	277 (90.5)	16 (5.2)	13 (4.3)
Smoking has the potential to contribute to the development of low back pain.	105 (34.3)	114 (37.3)	87 (28.4)
Educational level has an impact on the development of low back pain.	122 (39.9)	48 (15.7)	136 (44.4)
Marital status is a risk factor for low back pain.	47 (15.4)	50 (16.3)	209 (68.3)
Practices on ergonomic factors contributing to LBP	Agree	Undecided	Disagree
	N (%)	N (%)	N (%)
Are you involved in heavy manual lifting of patients from floors and beds?	305 (99.7)	1 (0.3)	0
Do you lift and move equipment?	298 (97.4)	4 (1.3)	4 (1.3)
Do you position patients in bed?	288 (94.1)	14 (4.6)	4 (1.3)
Do you stand for long standing hours?	298 (97.4)	8 (2.6)	0
Do you bend while working?	303 (99.0)	3 (1.0)	0
Do you exert force while working?	294 (96.1)	9 (2.9)	3 (1.0)
Poor physical fitness	285 (93.1)	18 (5.9)	3 (1.0)
Are you working overloaded?	304 (99.4)	1 (0.3)	1 (0.3)
Is there work pressure?	296 (96.7)	8 (2.6)	2 (0.7)
Poor work environment	292 (95.4)	8 (2.6)	6 (2.0)
Support from managers	184 (60.1)	51 (16.7)	71 (23.2)
There is work control	202 (66.0)	61 (19.9)	43 (14.1)
Job satisfaction	172 (56.2)	59 (19.3)	75 (24.5)
Do you work fatigued?	277 (90.5)	9 (2.9)	20 (6.5)
Does your work result in emotional distress?	246 (80.4)	29 (9.5)	31 (10.1)

**Table 3 healthcare-13-00083-t003:** An association between knowledge, attitudes and perceptions, and respondent characteristics.

Characteristic	Knowledge			Attitude			Practice		
	High	Low	Total	Negative	Positive	Total	Poor	Good	Total
Age group (years) <30 30–40 >40	69 (73.4) 97 (66.4) 42 (63.6)	25 (26.6) 49 (33.6) 24 (36.4)	94 (30.7) 146 (47.7) 66 (21.6)	49 (52.1) 70 (48.0) 23 (34.9)	45 (47.9) 76 (52.1) 43 (65.2)	94 (30.7) 146 (47.7) 66 (21.6)	50 (53.2) 69 (47.3) 25 (37.9)	44 (46.8) 77 (52.7) 41 (62.1)	94 (30.7) 146 (47.7) 66 (21.6)
Total	208 (68.0)	98 (32.0)	306 (100.0)	142 (46.4)	164 (53.6)	306 (100.0)	144 (47.1)	162 (52.9)	306 (100.0)
	χ^2^ (df = 2) = 2.00, *p* = 0.368 (Non-significant association)			χ^2^ (df = 2) = 4.92, *p* = 0.085 (Non-significant association)			χ^2^ (df = 2) =3.65, *p* = 0.161 (Non-significant association)		
Gender Male Female	68 (72.3) 140 (66.0)	26 (27.7) 72 (34.0)	94 (30.7) 212 (69.3)	46 (48.9) 96 (45.3)	48 (51.1) 116 (54.7)	94 (30.7) 212 (69.3)	53 (56.4) 91 (42.9)	41 (43.6) 121 (57.1)	94 (30.7) 212 (69.3)
Total	208 (68.0)	98 (32.0)	306 (100.0)	142 (46.4)	164 (53.6)	306 (100.0)	144 (47.1)	162 (52.9)	306 (100.0)
	χ^2^ (df = 2) = 1.19. *p* = 0.276 (Non-significant association)			χ^2^ (df = 2) = 0.350, *p* = 0.554 (Non-significant association)			χ^2^ (df = 2) = 4.73, *p* = 0.030 (Significant association)		
Marital status Single Married Divorced	134 (70.2) 71 (63.4) 3 (100.0)	57 (29.8) 41 (36.6) 0	191 (62.4) 112 (36.6) 3 (1.0)	89 (46.6) 50 (44.6) 3 (100.0)	102 (53.4) 62 (55.4) 0	191 (62.4) 112 (36.6) 3 (1.0)	92 (48.2) 50 (44.6) 2 (66.7)	99 (51.8) 62 (55.4) 1 (33.3)	191 (62.4) 112 (36.6) 3 (1.0)
Total	208 (68.0)	98 (32.0)	306 (100.0)	142 (46.4)	164 (53.6)	306 (100.0)	144 (47.1)	162 (52.9)	306 (100.0)
	Fisher’s exact *p* = 0.275 (Non-significant association)			Fisher’s exact *p* = 0.195 (Non-significant association)			Fisher’s exact *p* = 0.634 (Non-ignificant association)		
Educational level Diploma Degree Masters	177 (68.6) 29 (64.4) 2 (66.7)	81 (31.4) 16 (35.6) 1 (33.3)	258 (84.3) 45 (14.7) 3 (1.0)	126 (48.8) 16 (35.6) 0	132 (51.2) 29 (64.4) 3 (100.0)	258 (84.3) 45 (14.7) 3 (1.0)	124 (48.1) 20 (44.4) 0	134 (51.9) 25 (55.6) 3 (100.0)	258 (84.3) 45 (14.7) 3 (1.0)
Total	208 (68.0)	98 (32.0)	306 (100.0)	142 (46.4)	164 (53.6)	306 (100.0)	144 (47.1)	162 (52.9)	306 (100.0)
	Fisher’s exact *p*= 0.824 (Non-significant association)			Fisher’s exact *p* = 0.062 (Non-significant association)			Fisher’s exact *p* = 0.322 (Non-significant association)		
How long have you been a nurse (years) <10 10–20 >20	96 (72.7) 82 (64.6) 30 (63.8)	36 (27.3) 45 (35.4) 17 (36.2)	132 (43.1) 127 (41.5) 47 (15.4)	69 (52.3) 56 (44.1) 17 (36.2)	63 (47.7) 71 (55.9) 30 (63.8)	132 (43.1) 127 (41.5) 47 (15.4)	69 (52.3) 62 (48.8) 13 (27.7)	63 (47.7) 65 (51.2) 34 (72.3)	132 (43.1) 127 (41.5) 47 (15.4)
Total	208 (68.0)	98 (32.0)	306 (100.0)	142 (46.4)	164 (53.6)	306 (100.0)	144 (47.1)	162 (52.9)	306 (100.0)
	χ^2^ (df = 2) = 2.42, *p* = 0.298 (Non-significant association)			χ^2^ (df = 2) = 4.08, *p* = 0.130 (Non-significant association)			χ^2^ (df = 2) = 8.70, *p* = 0.013 (Significant association)		

## Data Availability

Data from this study are the property of the Government of South Africa and the University of KwaZulu-Natal; thus, it cannot be made publicly available. All interested readers can access the data set from the Chairperson of the South African Health Research and Ethics Committee (BREC) using the following contacts: The Chairperson of the South Africa Health Research and Ethics Committee, Email: hrkm@kznhealth.co.za, Tel.: +27-(033)-395-2805. The Chairperson BIOMEDICAL RESEARCH ETHICS ADMINISTRATION Research Office, Westville Campus, Govan Mbeki Building University of KwaZulu-Natal, P/Bag X54001, Durban, 4000 KwaZulu-Natal, South Africa, Tel.: +27-31-260-4769, Fax: +27-31-260-4609, Email: BREC@ukzn.ac.za.

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
