# Peer review of "Ergonomics and Occupational Health: Knowledge, Attitudes and Practices of Nurses in a Tertiary Hospital in Botswana"

_healthcare, 2025, doi:10.3390/healthcare13010083_

Round 1

Reviewer 1 Report

Comments and Suggestions for Authors

Thank you for your work ,In general i think your work addressed the aim and your idea relevant to your field. In abstract presentation not need to add a lot of numbers .

in introduction, you should add some paragraph on self-administered questionnaire 

in methodology need to see flow chart 

Figure & table need more to be more concise

references were appropriate but need to be update 

Reviewer 2 Report

Comments and Suggestions for Authors

You conducted a study on nurses' knowledge, attitudes and practices regarding ergonomic principles. This study is an important issue in terms of preventing occupational injuries at work among nurses. Overall, the framework of the paper is well organized. However, in order to publish the paper, the following revisions are required.

. Please first explain the full term of PMH in line number 105.

. Why was Scoring the questions at line number 152 not numbered? Wouldn't it be a bit more comprehensive to specify 2.8 and call it Variables?

. Ethical consideration seems to need to be revised to 2.8.

. At the bottom of Results, separate each title with a number as shown in 3.1.

for example, 3.1. Characteristics of the study population

. Even in Discussion, separate each title with a number in front of it.

for example, 4.1. Demographic characteristics of participants

. The conclusion is very short. The conclusion should include the purpose of the study, a brief description of the methods, the main results, including numbers, and social or policy implications. Please correct and supplement.

Reviewer 3 Report

Comments and Suggestions for Authors

Please make a swot analysis of your study.

Also mention the paper below by comparing the gender and spinal curvature related issues, would it be any good if you were able to record the spinal curvatures during your studies and use a tool like this etc...

  1.  Spinal Curvature for the Assessment of Spinal Stability. Int. J. Biomed. Eng. Technol. 201620, 226. [Google Scholar] [CrossRef]

Reviewer 4 Report

Comments and Suggestions for Authors

The article presents interesting data on occupational safety in medical facilities. The presented results not only indicate but also underscore the urgent need to educate medical personnel on preventing occupational diseases, a crucial step in ensuring their health and safety.

The article is well-formulated. The content has been presented in proportion to the journal's editorial requirements. The methodology used in the research has been described in a clear and understandable way, instilling confidence in the study's findings. The extensive bibliography indicates in-depth research on the subject.

The abbreviations MSD and LBP are not explained in the article's main text; they are explained only in the abstract.

Reviewer 5 Report

Comments and Suggestions for Authors

The article is a report on research conducted among nurses in public tertiary hospitals in Botswana. This study aimed to assess knowledge, attitudes and practices of nurses towards ergonomic principles. This goal can be accepted, as long as such research was treated as preliminary. If the research ended with the questions presented, then in my opinion the entire commitment of the researchers and participants was wasted. At a small cost, the survey questionnaire could have been supplemented with questions about the use of knowledge, i.e. how the most burdensome activities are performed, and questions about the individual assessment of workload and/or fatigue, as well as questions about the frequency of typical work-related ailments. Such data would allow determining the effects of having or not knowing about ergonomic principles. Only analysis of such data would be helpful in creating programs for improving knowledge, skills and behaviors. In my opinion, the current research was conducted correctly, but the entire project is imperfect. Nevertheless, in my opinion, the article can be published, but on condition that the discussion is expanded to include the problems indicated above.

Reviewer 6 Report

Comments and Suggestions for Authors

Interesting topic, comments and small suggestions for improvement are made in the review report.

Comments on the Quality of English Language

See comments in the review report.

Round 2

Reviewer 2 Report

Comments and Suggestions for Authors

Thank you for your efforts.